# Intrinsic Robustness of Prophet Inequality to Strategic Reward Signaling

**Wei Tang**[*]     **Haifeng Xu**[†]     **Ruimin Zhang**[‡]     **Derek Zhu**[§]

## Abstract

Prophet inequality concerns a basic optimal stopping problem and states that simple threshold stopping policies — i.e., accepting the first reward larger than a certain threshold — can achieve tight $1/2$-approximation to the optimal prophet value. Motivated by its economic applications, this paper studies the robustness of this approximation to natural *strategic manipulations* in which each random reward is associated with a self-interested player who may selectively reveal his realized reward to the searcher in order to maximize his probability of being selected.

We say a threshold policy is $\alpha$(-strategically)-robust if it (a) achieves the $\alpha$-approximation to the prophet value for strategic players; and (b) meanwhile remains a $1/2$-approximation in the standard non-strategic setting. Starting with a characterization of each player's optimal information revealing strategy, we demonstrate the intrinsic robustness of prophet inequalities to strategic reward signaling through the following results: (1) for arbitrary reward distributions, there is a threshold policy that is $\frac{1-1/e}{2}$-robust, and this ratio is tight; (2) for i.i.d. reward distributions, there is a threshold policy that is $1/2$-robust, which is tight for the setting; and (3) for log-concave (but non-identical) reward distributions, the $1/2$-robustness can also be achieved under certain regularity assumptions.[5]

## 1   Introduction

The prophet inequality of Krengel and Sucheston [37] is a foundational framework for the theory of optimal stopping problems and sequential decision-making. In the classic prophet inequality, a searcher faces a finite sequence of non-negative, real-valued and independent random variables $X_1, \ldots, X_N$ with known distributions $H_i$ from which a reward of value $X_i$ is drawn sequentially for $i = 1, \ldots, N$. Once a random reward is realized, the searcher decides whether to accept the realized reward and stop searching, or reject the reward and proceed to the next reward. The searcher's objective is to maximize the value of the accepted reward. The performance of the searcher's stopping policy is evaluated against a *prophet value* which equals to the ex-post maximum realized reward. The classic and elegant result of Samuel-Cahn [42] showed that a simple static threshold policy achieves at least half of the prophet value, and surprisingly, this bound is the best possible even among dynamic policies. Samuel-Cahn's policy uses the threshold that is the median of the distribution of the highest prizes, and then accepts the first realized reward that exceeds this threshold. The existence of this $1/2$-approximation is now known as the *prophet inequality*.

Recently, there is a regained interest of the prophet inequality due to its beautiful connection to online mechanism design (see, e.g., [31, 16]), and many different settings of the prophet inequality

---

[*]Chinese University of Hong Kong. Email: `weitang@cuhk.edu.hk`.

[†]University of Chicago. Email: `haifengxu@uchicago.edu`.

[‡]University of Chicago. Email: `ruimin@uchicago.edu`.

[§]University of Chicago. Email: `dzhu8@uchicago.edu`.

[5]The full version of this paper can be found at https://arxiv.org/pdf/2409.18269

38th Conference on Neural Information Processing Systems (NeurIPS 2024).

have been studied (see the survey by [40, 19]). For example, Kleinberg and Weinberg [36] extend the prophet inequality to all matroid constraints and similar to [42], they also show that a threshold stopping policy with the threshold equal to half of the expected maximum reward can also lead to the optimal $1/2$-approximation. Indeed, it is now well-known that there exists a range of thresholds for the classic prophet inequality that can achieve the optimal $1/2$-approximation (see Definition 2.2).

An important assumption in the classic prophet inequality is that the distribution of each random variable is an inanimate object and, once the searcher reaches it, it will fully disclose its realized reward to the searcher. Yet this may not be the case in many real-world applications where each distribution may often be associated with a *strategic player*[6] who may have incentives to selectively disclose information to maximize his own probability of being chosen by the searcher. This is usually the case when information is not controlled by nature but by humans or algorithms. Such examples are ample in economic activities. For instance, when a recruiter searches for the best candidate for a position by sequentially interviewing a set of job applicants, each job applicant is naturally a strategic player and would want to control how much information they disclose about their characteristics (including strengths, weaknesses, personality, and experience) to the recruiter (the searcher) so as to maximize the probability of being hired. Similarly, when a venture capitalist searches for the most promising startup to invest by sequentially visiting each startup, the startups are strategic players. They have more accurate information about their own potential and can control how much of this information they disclose to attract the investment from the venture capitalist (the searcher). Finally, in the real estate market, when a buyer searches for the most attractive property to purchase, the property sellers are strategic players who can control the amount of information they disclose about the property's condition and potential returns to the buyer (the searcher).

Motivated by real-world applications like above, we introduce and study a natural variant of the prophet inequality where each reward distribution is associated with a strategic player who can decide what information about the realized reward will be disclosed to the searcher. To capture such partial information revelation, we adopt the standard framework of information design (also known as Bayesian persuasion [35, 10]) and assume that each player can selectively disclose reward information by implementing an information strategy – often referred to as an *experiment* or *signaling scheme* [35, 10] – which stochastically maps the realized reward (unobservable to the searcher) to a random signal (observable to the searcher). For the motivating applications above, such revelation strategies encode answers to standard interview questions (e.g., experiences, behavioral questions) prepared in advance by job candidates, the presentations prepared by startups for the VC, or the property's brochures prepared by property sellers. Each player aims to maximize his probability of being chosen by the searcher, leading to a (constant-sum) competition among them.

**Our Results.** In this work, we stand in the searcher's shoes and look to understand how robust the classic prophet inequalities are to the above strategic player behaviors. Like most previous work in this space, we restrict our attention to threshold stopping policies. We start by characterizing players' optimal information revealing strategies. Given any threshold stopping policy with threshold $T$, the optimal information strategy of player $i$ with prior reward distribution $H_i$ has the following clean threshold structure (albeit using a threshold different from $T$): there exists a reward cutoff $t_i$ such that player $i$ simply reveals whether $X_i \geq t_i$ or $X_i < t_i$. Moreover, $t_i$ satisfies $\mathbb{E}[X_i | X_i \geq t_i] = T$. In words, player $i$ simply pools all the "good rewards" together to make their expectation barely pass the threshold $T$.[7] This characterization allows us to reduce players' strategic behaviors to a related prophet inequality problem with binary reward supports. Our later analysis hence only needs to further investigate how well the thresholds for classic prophet inequalities perform in this (related) additional problem.

Armed with the above characterization, next we turn to analyze the intrinsic robustness of the classic prophet inequality. We say a threshold stopping policy with threshold $T$ is $\alpha$(-strategically)-robust if it retains the $1/2$-approximation in the classic setting and meanwhile also can achieve $\alpha$-approximation in the strategic scenario when all the players optimally reveal information. Our first main result is that, for arbitrary reward distributions, the threshold policy with $T$ equaling the *half expected max* threshold of Kleinberg and Weinberg [36] is $\frac{1-1/e}{2}$-robust (Theorem 4.1). Moreover,

---

[6]We refer to the one that controls the reward information as the "player", and refer to the one that decides when to stop as the "searcher".

[7]More formally, these response strategies form a subgame perfect Nash equilibrium among the sequential-move game of players (which turns out to be unique as far as players' utilities are concerned) (see Remark 3.2).

this competitive ratio of $\frac{1-1/e}{2} \approx 0.316$ is the best among all known threshold policies that can secure the $1/2$-approximation in the classic non-strategic setting (Proposition 4.2). This suggests that this well-studied threshold policy can also perform robustly well even when players are all strategic, illustrating the intrinsic robustness of the classic prophet inequality.

When the reward distributions are identical – referred to as *IID distributions* which have been extensively studied in literature [41, 18, 6, 30, 1, 33] – we show that there exists a threshold in the range of known thresholds (Definition 2.2) that is $1/2$-robust (Theorem 5.1). Moreover, this $1/2$ approximation ratio is optimal among all possible threshold stopping policies for any IID distributions (Proposition 5.2). Finally, when the reward distributions are not identical but log-concave, under certain regularity conditions we show that any threshold between the expected max and the median of the highest reward [42] is $1/2$-robust (Theorem 5.4). Note that log-concave reward distributions have been considered in previous works on prophet inequality (see, e.g., [6, 18]); moreover, it is satisfied by a wide range of distributions (e.g., normal, uniform, Gamma, Beta, Laplace, etc., [8]) and is a widely adopted assumption in algorithmic game theory (see, e.g., [15]).

**Additional Related Work.** Prophet inequality is a fundamental problem in optimal stopping theory which was introduced in the 70s [38, 37]. Recently there has been a growing interest in prophet inequalities, generalizing the problem to different settings [2, 7, 18, 21, 22, 36, 17, 4, 6, 20]. Our discussion here cannot do justice to its rich literature; hence, we refer interested readers to the recent survey by Lucier [40], Correa et al. [19] for comprehensive overviews of recent developments and its connections to economic problems. This work takes an informational perspective and associate each distribution in the prophet inequality with a strategic player that strategically signals reward information to the searcher. We follow the information design literature [35, 10] and allow players to design signaling schemes to influence the searcher's beliefs about the realized rewards. In our considered game, players are competing with each other for the selection of the searcher. The players' game thus also shares similarity with competitive information design [26, 28, 5, 23, 34].

Conceptually, our work also relates to the rapidly growing literature on strategic machine learning (see, e.g., [45, 32, 29, 24, 43, 46, 9]), which studies learning from strategic data providers. We also study similar strategic reward providers, albeit in a different algorithmic problem, i.e., optimal stopping. More generally, our work subscribes to the literature on information design in sequential decision-making. In particular, our paper relates to the recently increased interests on using online learning approaches to study the regret minimization when an information-advantaged player repeatedly interacts with an information-disadvantaged player [11, 14, 25, 44, 47] without knowing their preferences. Instead of focusing on regret minimization, we use the competitive ratio (as conventionally used in prophet inequalities) to measure the searcher's policy. It is worth mentioning that [30] also studies strategic information revealing in prophet inequality problems. In their setting, a centralized player strategically discloses reward information to the searcher, while players in our setting form a decentralized game and each player acts on his own to maximize his payoff.

## 2 Preliminary

In this section, we first revisit the formulation of the classic prophet inequality problem, and then formally introduce our setting as its natural variant where each distribution is associated with a strategic player who will be strategically signaling their reward information.

**Classic prophet inequality.** In standard settings, a searcher faces a finite sequence of known distributions $H_{1:N} \triangleq (H_i)_{i \in [N]}$ of $N$ non-negative independent random variables. The outcomes (i.e., the rewards) $X_i \sim H_i$ [8] for $i \in [N]$ are revealed sequentially to the searcher for $i = 1, \ldots, N$. Let $\lambda_i \triangleq \mathbb{E}_{H_i}[X_i]$ denote the mean reward for distribution $H_i$. Upon seeing a reward, the searcher decides whether to accept the observed reward and stop the process, or irrevocably move on to the next reward. The searcher's goal is to maximize the expected accepted reward. The searcher's expected payoff cannot be more than that of a prophet who knows in advance the realizations of the rewards, namely, $X_1, \ldots, X_N$. We denote by $\mathsf{OPT} \triangleq \mathbb{E}_{H_1,\ldots,H_N}\left[\max_{i \in [N]} X_i\right]$ the prophet value. Given a stopping policy $q$, let $X^{(q)}$ be the accepted reward. The stopping policy $q$ is said to be $\alpha$-approximate (of the prophet value) if the following holds for the searcher's expected payoff for

---

[8]Here $H_i$ denotes the cumulative distribution function (CDF) of the reward $X_i$

any input distributions $H_{1:N}$:

$$\mathbb{E}\left[X^{(q)}\right] \geq \alpha \cdot \mathsf{OPT} .$$

The above statement is often referred to as the prophet inequality. For any stopping policy $q$, the largest constant $\alpha \in (0, 1]$ that satisfies the above inequality is named the competitive ratio. Classic results of [38, 37] elegantly show that there exist simple threshold stopping policies that achieve the competitive ratio $1/2$, and moreover the ratio of $1/2$ is tight.[9]

**Definition 2.1** (Threshold Stopping Policy). *A threshold stopping policy is an online algorithm which pre-computes a threshold value $T$ as a function of the distributions $H_{1:N}$, and then accepts the first reward $X_i$ whose value is no smaller than the threshold, i.e., $X_i \geq T$.*

It is well-known that multiple thresholds can achieve the optimal $1/2$-approximation ratio.

**Definition 2.2** ($1/2$-approximation Threshold Spectrum). *Let $T_{\mathsf{KW}} \triangleq \mathbb{E}[\max_i X_i]/2$ (Kleinberg and Weinberg [36]), let $T^*$ satisfy $T^* = \sum_{i \in [N]} \mathbb{E}[(X_i - T^*)^+]$, and let $T_{\mathsf{SC}} \triangleq \sup\{t : Pr[\max_i X_i \geq t] \geq 1/2\}$ (Samuel-Cahn [42]). Then any threshold stopping policy with threshold $T \in [T_{\mathsf{KW}}, \max\{T_{\mathsf{SC}}, T^*\}]$ guarantees $1/2$-approximation to the prophet value.[10]*

**Prophet inequality with strategic reward signaling.** In the strategic setting, each distribution $H_i$ may be associated with a strategic player that governs how much information about the realized reward he would like to reveal to the searcher once the searcher reaches his distribution. Formally, upon arriving at the reward distribution $H_i$, the searcher does not directly observe the realized reward $X_i \sim H_i$. Instead, she observes an information signal, designed by the player $i$ with distribution $H_i$, that is correlated with the reward $X_i$. We follow the literature in information design [35] to model this strategic reward signaling: Each player $i$ chooses a signaling scheme $\phi_i(\cdot \mid x) \in \Delta(\Sigma_i)$, where $\Sigma_i$ is a measurable signal space and $\phi_i(\sigma \mid x) \in [0, 1]$ specifies the conditional probability of a signal $\sigma \in \Sigma_i$ that will be sent to the searcher when the reward $X_i = x \sim H_i$ is realized. Notice that, upon seeing a signal $\sigma \sim \phi_i(\cdot \mid x)$, together with the prior information $H_i$ from which the reward is realized, the searcher can update her Bayesian belief about the underlying realized value $X_i$, and then decides whether to stop and choose player $i$, or reject $i$ to continue her search.

Each player $i$ is competing with each other for the final selection from the searcher. Namely, a player obtains payoff 1 if his reward is accepted and payoff 0 if his reward is not accepted.[11] Each player's goal is to design an information revealing strategy that maximizes his probability of being chosen by the searcher. We below give two simple examples of information revealing strategies.

**Example 2.3** (Examples of Information Strategies). *(1) No information strategy: the signal is completely uninformative (e.g., the distribution $s\phi_i(\cdot \mid x)$ is a Dirac delta function on a single signal, i.e., $|\Sigma_i| = 1$), hence the searcher infers an expected reward of $\lambda_i = \mathbb{E}[X_i]$ as her perceived reward from player $i$; (2) Full information revealing strategy: the signal perfectly reveals player $i$'s value to the searcher (i.e., $\phi_i(\sigma \equiv x \mid x) = 1$ for every realized $X_i = x$, and $\Sigma_i = \mathsf{supp}(H_i)$)*

In this work, standing in the searcher's shoes, we are interested in how threshold stopping policies perform under players' optimal strategic reward signaling.

**Game Timeline.** The timeline of our prophet inequality with strategic players problem, where the searcher employs a threshold stopping policy, can be detailed as follows: (1) Knowing $H_{1:N}$, the searcher first announces a threshold stopping policy with threshold $T$ that is a function of $H_{1:N}$; (2) Knowing threshold $T$, each player then picks a signaling scheme (also known as an *experiment* in economics literature [35, 10]) to reveal partial information about the underlying reward; and (3) The searcher learns all players' information strategies, and then conducts a search based on her threshold

---

[9]See [40, 19] for the hardness example for the $1/2$-approximation.

[10]There also exists other thresholds that could lead to $1/2$-approximation, e.g., any $T \in [\min\{\underline{T}_{\mathsf{SC}}, T_{\mathsf{KW}}\}, T_{\mathsf{KW}})$ where $\underline{T}_{\mathsf{SC}} \triangleq \inf\{t : Pr[\max_i X_i \geq t] \geq 1/2\}$. However, using these thresholds requires to modify the policy defined in Definition 2.1 to be a *strict* stopping policy to obtain $1/2$-approximation (i.e., searcher accepts a first reward that is strictly larger than threshold). Moreover, under strict stopping policy, there may exist no Nash equilibrium among players' game that we formulate shortly.

[11]All of our results hold if each player $i$ has payoff $v_i$ if his reward is accepted and payoff $u_i$ if his reward is not accepted as long as $v_i > u_i$.

stopping policy with threshold $T$ (i.e., accepting the first player whose posterior mean of his reward distribution, given his revealed information signal, exceeds the threshold $T$). The assumption that the searcher knows information strategies as well as realized signals is commonly adopted in the information design literature [35, 10], and is also well motivated for the domains of our interest. For instance, when startups persuade VCs or property sellers persuade buyers, the signaling scheme could correspond to startups' product exhibitions or sellers' promotion brochures which determine what information the searcher could see. Misreporting realized signals corresponds to revealing untrue information, which not only violates regulation policies and but also causes the players to lose credibility in the long term.

Slightly abusing the notation, we also use $X^{(q)}$ to denote the accepted reward given the searcher's stopping policy $q$ under the strategic reward signaling, and let $u^{\mathsf{s}}(q) \triangleq \mathbb{E}\big[X^{(q)}\big]$ be the searcher's expected payoff. Notice that here the expectation is not only over the randomness of the distributions $H_{1:N}$, but also the information strategies $\{\phi_i(\cdot \mid x)\}$. Anticipating the players' strategic behavior, the searcher wants a stopping policy that can still guarantee a good performance competing against the prophet value OPT.

As mentioned earlier, we are particularly interested in how previously studied threshold stopping policies described in Definition 2.2 perform under the players' strategic reward signaling. To formalize our goal, we introduce the following notion of strategic robustness.

**Definition 2.4** ($\alpha$(-strategically)-robust Stopping Policies)**.** *A stopping policy $q$ is $\alpha$-robust if it achieves $\alpha$-approximation to the* OPT *when players are strategically signaling their rewards, and it remains a $1/2$-approximation in the standard non-strategic setting.*

## 3 Warm-up: Characterizing Optimal Information Revealing Strategy

We start our analysis by showing that when the searcher adopts a threshold stopping policy (Definition 2.1), each player's optimal information revealing strategy admits clean characterizations.

**Proposition 3.1** (Optimal Information Revealing Strategy)**.** *Given a threshold stopping policy as in Definition 2.1 with threshold $T$, for each player $i$:*

- *if $T \leq \lambda_i$, then player $i$'s optimal information revealing strategy is the no information strategy;*
- *if $T > \lambda_i$, then player $i$'s optimal information revealing strategy is* threshold signaling *and determined by a cutoff $t_i$ that satisfies $T = \mathbb{E}[X_i | X_i \geq t_i] = \int_{t_i}^{\infty} x \, \mathrm{d}H_i(x)/(1 - H_i(t_i))$. That is, player $i$'s optimal signaling scheme sends one of two signals: $X_i \geq t_i$ or $X_i < t_i$.*[12]

We highlight the intuition behind Proposition 3.1 below. Under a threshold stopping policy, every player maximizes his utility by maximizing the probability that the signal's posterior expected reward is at least $T$ (hence is selected). When the stopping threshold $T \leq \lambda_i$, this probability is 1 and hence maximized when player $i$ simply reveals no information. When $T > \lambda_i$, this probability is maximized when player $i$ blends the highest rewards together to form a posterior mean just equal to $T$, which is exactly the scheme described in Proposition 3.1.

**Remark 3.1** (Addressing Point Masses)**.** *For ease of presentation, Proposition 3.1 assumes the distribution $H_i$ is continuous. When $H_i$ has point masses, Proposition 3.1 still holds, but with a more subtle description of the pooling cutoff $t_i$. We provide more detailed discussions in the full version of the paper.*

**Remark 3.2.** *In game-theoretic terminology, Proposition 3.1 characterizes the subgame Perfect Nash equilibrium (SPNE) for the multi-player sequential game induced by any threshold stopping policy that the searcher commits to. This SPNE happens to enjoy simple structures; indeed, each player's equilibrium strategy is only a function of the threshold $T$ and not on other players' strategies or their order. This clean characterization is a consequence of the simple structure of (static) threshold policies. If the searcher's stopping policy is instead dynamic (i.e., allowing the decision of player $i$ to depend on previously realized rewards), SPNE is also well-defined but will adopt significantly more complex structures. While analyzing the SPNE under these dynamic policies may also be interesting, it is beyond the scope of this work since our focus is to study the power of static*

---

[12]In the corner case when $T$ is larger than the upper bound of player $i$'s distribution, player $i$ will never get chosen and thus they have no strategy

*threshold policies, which is also a central theme in the study of prophet inequalities. Finally, we note that the optimal strategies characterized in Proposition 3.1 are not unique but they all result in the same utility for every player. This is because player $i$ is actually indifferent on how to disclose the reward when $X_i \leq t_i$, leading to many different but utility-equivalent information strategies.*

**Equivalent Representation of Optimal Information Revealing Strategy.** When $T > \lambda_i$, player $i$'s optimal information revealing strategy described in Proposition 3.1 pools all the rewards $X_i \sim H_i$ above $t_i$ together to forge a conditional mean value of $T$, and pools the remaining smaller rewards into another signal. We hence also refer to $t_i$ as the pooling cutoff. This threshold signaling scheme can be equivalently represented as a binary-support distribution $G_i$ supported on two realizations corresponding to the two signals respectively:

$$\Pr_{x \sim G_i}[x = T] = 1 - H_i(t_i), \quad \Pr_{x \sim G_i}[x = a_i] = H_i(t_i) \text{ where } a_i \triangleq \frac{\lambda_i - T(1 - H_i(t_i))}{H_i(t_i)} . \quad (1)$$

In the literature of information design, this distribution $G_i$ is also known as a mean-preserving contraction of prior reward distribution $H_i$ [27, 12, 13].

Viewing the player $i$'s optimal information strategy as the distribution $G_i$, one can simplify the interaction between the searcher and player $i$ as the following when the searcher visits player $i$: a random reward $X_i \sim G_i$ is realized, and the searcher stops if and only if $X_i \geq T$. With this observation, one can also reduce the original interaction to the following simplified protocol: (1) the searcher first decides a stopping threshold $T$; (2) each player $i$ chooses the distribution $G_i$ as in Equation (1) according to his prior $H_i$ and the threshold $T$;[13] and (3) the searcher visits each $G_i$ sequentially and stops when the first realized reward $X_i \geq T$ where $X_i \sim G_i$.

We emphasize that in the above reduction, given a threshold stopping policy with threshold $T$, the searcher's expected payoff under the strategic reward signaling can be computed as $u^{\mathsf{s}}(T) = \mathbb{E}\left[X^{(q)}\right]$ where the expectation is over distributions $G_{1:N}$ and $X^{(q)}$ is the first realized $X_i \sim G_i$ such that $X^{(q)} \geq T$.

# 4  Achieving $\frac{1-1/e}{2}$-robustness for Arbitrary Distributions

In this section, we show that for any distributions $H_{1:N}$, there exists a $\frac{1-1/e}{2}$(-strategically)-robust threshold stopping policy using a threshold within the spectrum in Definition 2.2. The main result in this section is stated below.

**Theorem 4.1.** *For any distributions $H_{1:N}$, a threshold stopping policy with threshold $T = T_{\mathsf{KW}}$ is $\frac{1-1/e}{2}$-robust.t*

From Definition 2.2, we know that using the threshold stopping policy with threshold $T_{\mathsf{KW}}$ can achieve the optimal $1/2$-approximation in the classic prophet inequality for any distributions $H_{1:N}$. Theorem 4.1 above shows another desired property of the threshold $T_{\mathsf{KW}}$: it can achieve $\frac{1-1/e}{2}$-approximation even when distributions $H_{1:N}$ are strategically signaling their rewards, thus establishing its $\frac{1-1/e}{2}$-robustness. Given the optimality of the threshold $T_{\mathsf{KW}}$ in the non-strategic setting, it would be intriguing to ask whether this threshold $T_{\mathsf{KW}}$ can also achieve $1/2$-approximation under the strategic setting, or if there exists a threshold within the spectrum in Definition 2.2 that can achieve $1/2$-approximation under the strategic setting. Below we show that the answer is No. In fact, any threshold stopping policy using a threshold from the spectrum in Definition 2.2 cannot achieve $(\frac{1-1/e}{2} + \varepsilon)$-approximation for any $\varepsilon > 0$ under strategic reward signaling.

**Proposition 4.2** (Tightness of Theorem 4.1). *There exist distributions $H_{1:N}$ such that no threshold from the spectrum in Definition 2.2 can achieve $\alpha$-robustness where $\alpha > \frac{1-(1-1/(N-1))^{N-1}}{2}$.*

Notice that $\lim_{N \to \infty} \frac{1-(1-1/(N-1))^{N-1}}{2} = \frac{1-1/e}{2}$. Thus, the above Proposition 4.2 establishes the tightness of the results in Theorem 4.1.

**Remark 4.1.** *We point out a subtlety in the above lower bound, which leads to an intriguing open problem. Proposition 4.2 shows that any threshold within the spectrum in Definition 2.2 cannot*

---

[13]When $T \leq \lambda_i$, distribution $G_i$ is a point mass at mean value $\lambda_i$.

*achieve $(\frac{1-1/e}{2} + \varepsilon)$-approximation under strategic reward signaling. However, this does not rule out the possibility of having a threshold outside that spectrum that achieves $1/2$-robustness. This is an interesting open question to resolve, though necessarily challenging since it is even already quite non-trivial to prove $1/2$-approximation in the non-strategic setting for thresholds outside the spectrum of Definition 2.2, let alone achieving $1/2$-approximation simultaneously in both worlds.*

Theorem 4.1 holds for all distributions, regardless of being discrete or continuous. In the remainder of this subsection, we present the proof of Theorem 4.1 only for the continuous distribution case. The proof of this case carries our core ideas but is cleaner to present.

## 4.1 (Partial) Proof of Theorem 4.1: The Case of Continuous Distributions

Our proof starts by upper bounding the prophet value OPT.[14]

**Lemma 4.3** (Upper Bounding OPT via Pooling Cutoffs). *Given a threshold stopping policy with threshold $T$, let $t_i$ be the pooling cutoff for each player $i$ defined as in Proposition 3.1. Let $I \triangleq \arg\max_{i \in [N]} t_i$, then we have $\mathsf{OPT} \leq H_I(t_I)t_I + \sum_i T(1 - H_i(t_i))$.*

*Proof of Lemma 4.3.* Let us fix a threshold $T$, and let $t_i$ be the pooling cutoff for player $i$ defined in Proposition 3.1. Define $b_i \triangleq (X_i - t_i)^+$. By definition, for each $i$, we have $X_i \leq t_i + b_i$. Thus,

$$
\begin{aligned}
\mathsf{OPT} = \mathbb{E}_{H_{1:N}}\left[\max_i X_i\right] &\leq \max_i t_i + \sum_i \mathbb{E}_{H_i}[b_i] \\
&\overset{(a)}{\leq} \max_i t_i + \sum_i (T - t_i) \cdot (1 - H_i(t_i)) \\
&\overset{(b)}{\leq} t_I H_I(t_I) + \sum_i T(1 - H_i(t_i)) ,
\end{aligned}
$$

where inequality (a) follows from the definition of pooling cutoff $t_i$ in Proposition 3.1 and inequality (b) follows from the definition of $I$, $t_i \geq 0$, and rearranging the terms. $\square$

With the above upper bound of prophet value, we have the following results:

**Lemma 4.4.** *Let $T^\dagger$ satisfy $\prod_{i=1}^N H_i(t_i^\dagger) = (\frac{N-1}{N})^N$ where $t_i^\dagger$ is defined in Proposition 3.1 with threshold $T^\dagger$, then searcher's expected payoff $u^{\mathsf{s}}(T^\dagger) \geq T^\dagger \left(1 - \prod_i H_i(t_i^\dagger)\right) \geq \frac{1-(\frac{N-1}{N})^N}{2} \cdot \mathsf{OPT}.$*[15]

*Proof of Lemma 4.4.* Given a stopping threshold $T$, by Proposition 3.1, we can lower bound the searcher's expected payoff as follows: $u^{\mathsf{s}}(T) \geq T \cdot \left(1 - \prod_{i=1}^N H_i(t_i)\right)$. Thus, together with

---

[14]All our analysis in the main text implicitly consider the case where $\max_i \lambda_i < \mathsf{OPT}/2$. If we have $\max_i \lambda_i \geq \mathsf{OPT}/2$, then the searcher can simply choose the threshold $T = \mathsf{OPT}/2$ which lies in the range of the thresholds defined in Definition 2.2 to obtain payoff $u^{\mathsf{s}}(T) \geq \mathsf{OPT}/2$. To see this, by Proposition 3.1, if $\max_i \lambda_i \geq \mathsf{OPT}/2 = T$, then there exists at least one player $j$ whose $\lambda_j \geq \mathsf{OPT}/2$ will choose the no information revealing strategy, and the searcher surely obtains a payoff no smaller than $\mathsf{OPT}/2$.

[15]Note that for non-continuous distributions $T^\dagger$ always exists but is defined a bit differently, please see the full version of the paper for more details.

Lemma 4.3, we have

$$\frac{u^{\mathsf{s}}(T)}{\mathsf{OPT}} \geq \frac{T \cdot \left(1 - \prod_{i=1}^{N} H_i(t_i)\right)}{\mathsf{OPT}}$$

$$\overset{(a)}{\geq} \frac{T \cdot \left(1 - \prod_{i=1}^{N} H_i(t_i)\right)}{t_I H_I(t_I) + \sum_i T(1 - H_i(t_i))}$$

$$\overset{(b)}{\geq} \frac{1 - \prod_{i=1}^{N} H_i(t_i)}{H_I(t_I) + \sum_i (1 - H_i(t_i))}$$

$$\overset{(c)}{\geq} \frac{1 - \prod_{i=1}^{N} H_i(t_i)}{N + 1 - \sum_i H_i(t_i)}$$

$$\overset{(d)}{\geq} \frac{1 - \prod_{i=1}^{N} H_i(t_i)}{N + 1 - (\prod_{i=1}^{N} H_i(t_i))^{\frac{1}{N}}} \, ,$$

where inequality (a) is by Lemma 4.3; inequality (b) is by the fact that $t_i \leq T$ for all $i \in [N]$; inequality (c) is due to $H_I(t_I) \leq 1$; and inequality (d) is by the AM-GM inequality. Now consider the function $f(x) \triangleq \frac{1-x}{N+1-Nx^{\frac{1}{N}}}$ over $x \in [0,1]$. By choosing $x^\dagger = (\frac{N-1}{N})^N$, we have $f(x^\dagger) = \frac{1-(\frac{N-1}{N})^N}{2}$. This implies that we have $T^\dagger \left(1 - \prod_i H_i(t_i^\dagger)\right) \geq \frac{1-(\frac{N-1}{N})^N}{2} \cdot \mathsf{OPT}$. $\qquad\square$

With the above Lemma 4.4, we are now ready to prove Theorem 4.1:

*Proof of Theorem 4.1.* From Lemma 4.4, we showed

$$T^\dagger \left(1 - \prod_i H_i(t_i^\dagger)\right) \geq \mathsf{OPT} \cdot \frac{1 - (\frac{N-1}{N})^N}{2} \, ,$$

where $t_i^\dagger$ is the pooling cutoff defined in Proposition 3.1 with the threshold $T^\dagger$. By definition of $T^\dagger$, we have $\prod_i H_i(t_i^\dagger) = (\frac{N-1}{N})^N \leq {}^1/e$. Thus we can deduce that $T^\dagger \geq {}^{\mathsf{OPT}}/2 = T_{\mathsf{KW}}$. Now let $t_i^\ddagger$ be the pooling cutoff when the searcher uses the stopping threshold $T_{\mathsf{KW}}$. Then we have

$$u^{\mathsf{s}}(T_{\mathsf{KW}}) \geq T_{\mathsf{KW}} \cdot \left(1 - \prod_i H_i(t_i^\ddagger)\right) \overset{(a)}{\geq} T_{\mathsf{KW}} \cdot \left(1 - \prod_i H_i(t_i^\dagger)\right) \geq \mathsf{OPT} \cdot \frac{1 - {}^1/e}{2}$$

where inequality (a) is due to $T^\dagger \geq T_{\mathsf{KW}}$, and thus we have $t_i^\dagger \geq t_i^\ddagger$. $\qquad\square$

# 5 Achieving $\frac{1}{2}$-robustness for Special Distributions

The preceding section showed the ${}^{(1 - {}^1/e)}/2$-robustness of the $T_{\mathsf{KW}}$-threshold stopping policy for arbitrary reward distributions. In this section, we show that this ratio can be improved to ${}^1/2$-robustness when the distributions $H_{1:N}$ satisfy certain conditions: (1) IID distributions – all reward distributions are identical, namely, $H \equiv H_i$ for all $i \in [N]$ (see Section 5.1); and (2) Log-concave distributions – reward distribution $H_i$ has log-concave density (see Section 5.2). We also show that ${}^1/2$-robustness is tight under IID distributions.

## 5.1 IID Distributions

Our main findings for IID distributions are stated below:

**Theorem 5.1.** *For any distributions $H_{1:N}$ where $H \equiv H_i, \forall i \in [N]$, a threshold stopping policy with threshold $T = T^*$ is $\frac{1}{2}$-robust where $T^*$ is defined in Definition 2.2.*

For IID distributions, we show that the searcher is able to achieve a better robustness approximation ratio compared to arbitrary distributions. Below we argue that this ${}^1/2$-robustness is tight in the sense that there exists no threshold policy that can achieve better robustness approximation ratios.

**Proposition 5.2** (Tightness of Theorem 5.1). *There exist IID distributions such that there exists no threshold stopping policy that can achieve $\alpha$-robustness where $\alpha > \frac{1}{2} + \varepsilon$ for any $\varepsilon > 0$.*

We note that Proposition 5.2 is a slightly stronger lower bound than Proposition 4.2 as it rules out the possibility for *all* possible threshold stopping polices, being within the spectrum in Definition 2.2 or beyond. We prove Proposition 5.2 by constructing a hard instance. In particular, we construct IID distributions with binary support. With this instance, we show that any threshold policy that achieves competitive ratio at least $1/2$ in non-strategic setting will have competitive ratio approaching to 0 in the strategic setting. Please see

A crucial requirement in our robustness study so far is that we insist that the threshold policy should, first of all, remains an $1/2$-approximation in the non-strategic setting[16], conditioned on which we look for additional guarantee for in the strategic setting. We conclude this section by pointing that if one was willing to give up the $1/2$-approximation in the non-strategic setting, then it is indeed possible to have a threshold policy that achieves better approximation (specifically, an $(1 - 1/e)$-approximation) in the strategic setting. While this does not satisfy our robustness requirement, it is useful to note.

**Corollary 5.3.** *For any IID distributions, there exists a threshold stopping policy that is $(1 - 1/e)$-approximation under strategic reward signaling. Moreover, there exist IID distributions such that no threshold stopping policy can achieve $(1 - 1/e + \varepsilon)$-approximation for any $\varepsilon > 0$.*

## 5.2 Log-concave Heterogeneous Distributions

In this subsection, we show that when the distributions $H_{1:N}$ satisfy certain regularity assumptions, there exist threshold stopping policies with thresholds from Definition 2.2 that can also achieve $1/2$-robustness. The main result in this section is stated as follows:

**Theorem 5.4.** *For $\alpha, \beta > 0$, let $F_{\alpha,\beta}$ be the family of distributions with log-concave probability density functions $f$ on support $[0,1]$ such that $f(1) \geq \alpha$ and $f'(1) \geq -\beta$. If the distributions $H_1, \ldots, H_N$ are all from $F_{\alpha,\beta}$ and $N \geq 1 + \frac{\beta}{\alpha^2}$, then we always have $2 \cdot T_{\mathsf{KW}} \leq T_{\mathsf{SC}}$ and any threshold $T$ satisfying $2 \cdot T_{\mathsf{KW}} \leq T \leq T_{\mathsf{SC}}$ is $1/2$-robust.*

A few remarks on the assumptions in Theorem 5.4 are worth mentioning. First, log-concavity of probability density functions[17] is a commonly used assumption; they include but are not limited to: normal, beta, gamma, and exponential distributions. The restriction to support on $[0,1]$ is for normalization reasons, hence without loss of generality. The main non-trivial restriction is that this result holds when the number of players $N$ is large enough, formally $N \geq 1 + \frac{\beta}{\alpha^2}$. This condition becomes less restrictive when $\alpha$ (lower bounding $f(1)$) becomes larger and/or $\beta$ (upper bounding $-f'(1)$) becomes smaller. These together, intuitively, imply that $f$ decreases slowly within $[0,1]$.

Define $\bar{H}(x) \triangleq \prod_{i=1}^{N} H_i(x)$. Theorem 5.4 follows directly from the following Lemma 5.5 and Lemma 5.6.

**Lemma 5.5.** *If $\bar{H}$ is convex, then $2 \cdot T_{\mathsf{KW}} \leq T_{\mathsf{SC}}$ and any threshold stopping policy with the threshold $T$ satisfying $2 \cdot T_{\mathsf{KW}} \leq T \leq T_{\mathsf{SC}}$, where $T_{\mathsf{KW}}, T_{\mathsf{SC}}$ are defined as in Definition 2.1, is $1/2$-robust.*

**Lemma 5.6.** *If the distributions $H_1, \ldots, H_N$ are all from $F_{\alpha,\beta}$ and $N \geq 1 + \frac{\beta}{\alpha^2}$, then $\bar{H}$ is convex.*

## 6 Conclusion and Future Directions

In this paper, we study a variant of the prophet inequality problem where each random variable is associated with a strategic player who can strategically signal their reward to the searcher. We first fully characterize the optimal information strategy of each player, then we show the threshold stopping policies that can perform robustly well under both the strategic and non-strategic settings.

Our novel consideration of natural strategic manipulations in prophet inequalities open the door for many interesting future directions. First, it is interesting to see if we can improve the $\frac{1 - 1/e}{2}$-robustness guarantee for arbitrary reward distributions by using not commonly used thresholds outside the spectrum of Definition 2.2. This may be technically challenging since finding a threshold

---

[16]Note that in non-strategic setting, unlike the case for arbitrary distributions where there exists no policy that can achieve better than $1/2$-approximation, for IID distributions, $(1 - 1/e)$-approximation can be achieved by fixed threshold together with some careful probabilistic tie-breaking rule [3] (or 0.7451-approximation with an adaptive threshold policy [39]). However, in our work, we focus on fixed threshold policy with deterministic tie-breaking rule where $1/2$-approximation is still optimal for IID distributions.

[17]A probability density function $f : \mathbb{R} \to \mathbb{R}^+$ is log-concave if $\log(f)$ is concave.

outside this spectrum with optimal $1/2$-approximation for the classic non-strategic prophet inequality is already non-trivial. Another interesting direction is to go beyond threshold policies. That leads to a different research theme, not about robustness of the classic prophet inequality, but rather in the search of potentially much more complex best-of-both-world policies. Our preliminary result reveals that a dynamic threshold stopping policy (using different thresholds for different players) has the potential to help the searcher to achieve more than $\frac{1-1/e}{2}$-approximation under the strategic setting. However, it is still unclear whether $1/2$-robustness is achievable under threshold stopping policies, no matter whether it is static threshold or dynamic threshold. Finally, even if one only cares about the performance under the strategic reward signaling (and ignore the non-strategic world), it is still unclear which static threshold stopping policy achieves the highest competitive ratio. The authors in [20] study a model with a very similar mathematical structure and proved that this best upper bound is strictly less than $1/2$ in Section 4.1, but an interesting direction is improving this bound further. We note that all our constructed examples in the hardness results (e.g., Proposition 4.2 and Proposition 5.2) do not rule out the existence of such $1/2$-approximation threshold stopping policies.

**Acknowledgment.** Haifeng Xu is supported in part by the NSF Award CCF-2303372, Army Research Office Award W911NF-23-1-0030 and ONR Award N00014-23-1-2802. We thank the anonymous reviewers for constructive and helpful comments.

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
