# OpenReview forum: "Intrinsic Robustness of Prophet Inequality to Strategic Reward Signaling"
_NeurIPS.cc/2024/Conference — NeurIPS 2024 poster_

### Official Review · Reviewer_BRGn · 2024-07-09

**Soundness:** 3
**Presentation:** 4
**Contribution:** 3
**Rating:** 7
**Confidence:** 3

**Summary:**

This paper studies the robustness of threshold algorithms when involving strategic manipulations in the classic prophet inequality problem. Specifically, the paper considers the scenario when each random reward of $N$ variables is associated with a strategic player, who can commit to a signaling scheme before the search, and the searcher can only see the signal rather than the actual reward during the whole search. Each player wants to maximize the probability of being chosen by the searcher and, thus, would optimally choose the signaling scheme against the searcher's thresholding algorithm. The authors first characterize any player's optimal scheme, which has a quite simple form. Based on the result, the authors show that threshold $T_\mathsf{KW}$ is a $(1 - 1/e)/2$-approximation in this setting and is tight. For i.i.d. distributions, $T^*$ gives a $1/2$-approximation, and is tight. For log-concave distributions, a spectrum of thresholds also gives a $1/2$-approximation.

**Strengths:**

Personally, I like the idea of involving incentives in the classic online decision-making problems, and this work successfully gives such a trial. This work is well-motivated in the real-world scenarios for the prophet inequality problem, where the searched agents have incentives to manipulate their true rewards by giving signals. Consequently, this paper transfers the decision-making problem into a game. Also, thresholding algorithms are concise and known to be optimal for the original problem, and it is interesting to see that they can also work well with strategic players under the result that the SPE for these players is also concise. I like this paper's motivation and results, and they can be set as a basis for future work.

**Weaknesses:**

This paper leaves some major problems unanswered. For example, can the authors provide some preliminary thoughts if we only require a $\beta < 1/2$ approximation in the original world and search for the Pareto frontier of $(\alpha, \beta)$? That being said, I think the authors have already provided enough.

**Questions:**

Please see the weaknesses above.

**Limitations:**

The authors have addressed the limitations of their work.

---

> ### Author Rebuttal · Authors · 2024-08-06
>
> We sincerely thank the reviewer for the positive feedback! Please find below our response to your comment.
>
> $(\alpha, \beta)$ **Pareto frontier**
> We thank the reviewer for this very interesting comment. Characterizing the Pareto frontier of the approximation ratios requires substantial additional effort. So far we do not have a clear picture on how this Pareto frontier would look like, and we agree with the reviewer this is an interesting open question to explore as the future work.

---

> > ### Comment · Reviewer_BRGn · 2024-08-08
> >
> > I appreciate your response!

---

### Official Review · Reviewer_1242 · 2024-07-10

**Soundness:** 4
**Presentation:** 3
**Contribution:** 3
**Rating:** 6
**Confidence:** 3

**Summary:**

The paper studies a variant of the classical prophet inequalities problem, where "boxes" are now strategic agents who can pool their realized value into bins and only reveal to the decision maker which bin it falls into.  The expected value in that bin then plays the role of the realized value.  The authors study best-of-both-worlds threshold policies, which (1) achieve the tight 1/2 approximation ratio in the non-strategic setting, and (2) achieves a ratio of alpha in the strategic model.  The main results are (conditional) optimal policies in the general setting, the IID setting, and the log-concave priors setting.

**Strengths:**

The model appears novel and sensible, and the results are relatively complete.  It is nice to see best-of-both-worlds guarantees are possible in this model, which wasn't clear to me before the fact.

**Weaknesses:**

Parts of the model can be better justified (see detailed comments).  I'd also appreciate the results better if there were unconditional / tighter impossibility results.

**Questions:**

(also including detailed comments)

Line 70, threshold policies: is this without loss of generality?

Timeline paragraph: I feel this is a bit ambiguous.  In particular, since the searcher has first-order commitment power, essentially the model assumes the searcher only commits to policies that put a threshold on the posterior expected value (otherwise the searcher would do something like "if you don't reveal the full information, you are out" and we are back in the classical model).  I feel this part of the model could be better motivated.

Characterization of players' response: this is reminiscent of the response of value maximizing players in [19], which has a very similar structure.  I wonder if a formal connection can be made between the two models.

Remark 4.1: there is an upper bound (i.e., impossibility result) strictly smaller than 0.5 in [19], which might imply the impossibility of 1/2-robustness in your model, given the intuitive connection between the two models.  This would make your results a bit more complete.

Around line 368, dynamic threshold: I think you can get 1/2-approximation in the strategic setting?  Take any 1/2-approximation policy in the non-strategic setting.  Construct a new policy such that each player's response cutoff under the new policy is the same as the threshold of the old policy.  Then you can couple the two policies in the two worlds and argue they have exactly the same performance?

**Limitations:**

No concerns.

---

> ### Author Rebuttal · Authors · 2024-08-06
>
> We sincerely thank the reviewer for the positive feedback! Please find below our response to each of your comments.
>
> **The generality of threshold policies**
> The reason that we restrict our attention to the threshold policies is that they are the policies that achieve the classic prophet inequality in the non-strategic setting. Given that our main focus in this work is to understand the robustness of the classic prophet inequality, it is indeed without loss to focus on the threshold policies. We will make this point more clear in the revision.
>
> **Threatening policy**
> This is a great point and we also thought about it previously. However, we felt that the threatening policy is not natural because it is not a subgame perfect Nash, which is the standard solution concept in such sequential interactions. Our model can be viewed as an approximate version of subgame perfect Nash, where the first player (i.e. the searcher) plays an approximately optimal strategy from a simple policy space (the exact optimal stopping policy is  likely very complex). The motivation for adopting such approximately optimal yet simple policy naturally echoes the entire literature on prophet inequality.
>
> **Connections to [19]**
> We thank the reviewer for pointing out the connection to the [19]. It is indeed interesting to see that the player’s optimal information revealing strategy has the similar structure to the response of the value maximizing buyers studied in [19], especially given that the player in our setting and the buyer in [19] have two very different optimization problems. We will add a discussion on this connection in the revision.
>
> After going through the hard example provided in [19], we can conclude that the same example can also be used in our setting to show the best approximation ratio that one can achieve for a static threshold policy is strictly smaller than 0.5 under a strategic setting, implying that a $1/2$-robustness is impossible for static threshold policy. We appreciate the reviewer for bringing out this impossibility result in [19] to our attention, and we will for sure add this result!
>
> **½-approximation policy in strategic settings**
> The reviewer is correct that there always exists an ½ approximation policy for the strategic settings. Indeed, the reviewer’s proposed method for constructing such a policy is essentially the dynamic threshold stopping policy (using different thresholds for different players) that we mentioned in Line 368 about our preliminary result.

---

> > ### Author Response · Authors · 2024-08-12
> >
> > We sincerely thank the reviewer for the valuable comments/suggestions and hope that our response has effectively addressed your main questions/comments. If there are any lingering questions, we would be more than happy to address them.

---

> > > ### Comment · Reviewer_1242 · 2024-08-13
> > >
> > > Thank you for your helpful response.  I don't have further questions.

---

### Official Review · Reviewer_vri4 · 2024-07-11

**Soundness:** 3
**Presentation:** 3
**Contribution:** 2
**Rating:** 6
**Confidence:** 3

**Summary:**

The paper studies a variant of the prophet inequality modeled as a game between the reward holders (Players $1, \ldots, n$) and the searcher. In this model, each reward $X_i$ is sampled from a distribution $H_i$ that is known to the searcher. However, each player $i$ can choose not to reveal $X_i$, and instead only reveal a signal $\Phi(X_i)$ that provides partial information about $X_i$. The objective of each player $i$ is to maximize the probability of being selected by the searcher. On the other hand, the searcher's goal is to maximize the expected value that it selects, while maintaining a competitive ratio of $1/2$ in the case where all players fully reveal their prices, i.e. in the standard prophet inequality. The authors first characterize the optimal information-revealing strategy for the players, then present an algorithm that achieves a competitive ratio of $(1-1/e)/2$ in this strategic setting and $1/2$ in the standard setting. They also prove that this competitive ratio is the best possible. Finally, they give improved competitive ratios in the strategic setting with IID and with Log-concave Heterogeneous Distributions, also maintaining a competitive ratio of $1/2$ in the standard prophet inequality.

**Strengths:**

* The setting is motivated
* The paper is well-written and presents a good balance between technical proofs and high level intuitions
* The paper presents some interesting results.

**Weaknesses:**

* The optimal revealing strategy is only characterized when the searcher follows a threshold policy
* The proposed algorithm is only optimal among threshold policies with thresholds in the spectrum given in Definition 2.2, and not among all possible algorithms.
* The paper requires the algorithms to maintain a competitive ratio of $1/2$ if all the players reveal their reward values. While this is a perfectly reasonable assumption in the general setting, it makes much less sense in the case of IID rewards. The optimal competitive ratio in the prophet inequality with IID random variables is $0.745...$. Even if we restrict ourselves to threshold policies, the optimal competitive ratio is $1-1/e$. The reasonable constraint in the IID case would then be that the algorithm maintains a competitive ratio of $1-1/e$ if all the players reveal their rewards.
* The problem assumes that the searcher does not observe the values $(X_i)_i$ but only signals $(\Phi(X_i))_i$. The threshold policy should then be defined as selecting the first reward $X_i$ such that $(\Phi(X_i) \geq T)$ and not $(X_i \geq T)$.

Minor weaknesses:
* Line 138: be more precise in the definition of the competitive ratio: it is the largest constant $\alpha$ satisfying $E[X^{(q)}] \geq \alpha \text{OPT}$
* Maybe Definition 2.2 should be a Lemma or Proposition instead
* Remark 3.1 should mention that Proposition A.1 also addresses the case where $X_i < T$ almost surely, not only the case of distributions with point masses, otherwise $t_i$ does not look well-defined.

**Questions:**

* See weaknesses
* The game timeline (line 170) is confusing to me: the searcher selects a threshold without knowing the types of signals they will observe, and then the players select revealing schemes. How does the searcher choose its stopping strategy without even knowing the type of information they will observe, which can be for example binary ($\Phi(X_i) = (X_i > a)$), negative $\Phi(X_i) = -X_i$, constant $\Phi(X_i) = 1)$, or others ?

**Limitations:**

The assumptions for all the are clearly stated for all the claims made in the paper.

---

> ### Author Rebuttal · Authors · 2024-08-06
>
> We sincerely thank the reviewer for the feedback and valuable comments! We notice that there may exist some conceptual misunderstandings in the reviewer's comments about the searcher’s stopping policy and the reason for us to study threshold stopping policies. So we would like to first clarify these potential misunderstandings and then provide our response to the other comments.
>
> **Clarification of the searcher’s stopping policy**
> We kindly recall that $\phi_i(\cdot \mid X_i)$ denotes the probability distribution over the signal space when the player $i$’s realized reward is $X_i$. Here, the signal space could be an arbitrary measurable space. Thus, directly comparing the realized signal with the stopping threshold is not feasible..
>
> Indeed, the stopping threshold $T$ is used for the searcher to decide whether she should accept the realized reward or move to the next player. Yet in our setting, even though the searcher is not able to directly observe the realized reward, she can observe the realized signal. With the observed signal, the searcher is able to derive a Bayesian posterior belief over the underlying realized reward $X_i$. Then the stopping threshold $T$ is used for the searcher to decide whether she should accept the current player by looking at whether the mean of this posterior belief is larger or smaller than the stopping threshold $T$.
>
> For example, if the reward distribution is uniform over $[0, 1]$, and the signaling scheme is $\phi(X_i) = \Bbb{1}_{X_i > a}$ (i.e., the searcher can observe whether the realized reward is smaller or larger than $a$), and if the searcher observes a signal that tells \indicator{X_i > a} = 1, then the searcher is able to update her Bayesian posterior and know that the expected realized reward should equal to (1+a)/2. Consequently, the searcher’s decision should be based on comparing $(1+a)/2$ and the stopping threshold $T$.
>
> **Benchmarking within the threshold stopping policies**
> We would like to clarify that the primary goal of this work, as also apparent from the paper's title, is to understand whether the well-studied prophet inequality (which is achieved by the static threshold stopping policies with the thresholds defined in Definition 2.2) is robust to the players’ strategical reward signaling.
> Our motivation is to study how these threshold stopping policies would perform if the players are strategically disclosing reward information (also known as the “best-of-both-worlds” style of results). Characterizing the approximation ratio for general stopping policies is also an interesting question (with a different motivation), but is beyond the scope of this paper.
>
>
> **Optimal revealing strategy when the searcher follows a threshold policy**
> We would like to clarify that our characterized optimal information revealing strategy actually holds as long as the researcher uses a stopping policy that only depends on the reward distributions $ (H_i)_{i\in[N]}$. So it is beyond just "following a threshold policy" as the reviewer suggested.
> On the other hand, as we mentioned in Remark 3.2, when the searcher’s stopping policy is more generally allowed to even depend on the realized signal or the information strategies, the subgame Perfect Nash equilibrium among players’ game would adopt significantly more complex structures and it requires complex backward induction analysis.  While analyzing this challenging situation could also be a technically interesting question, it deviates from our current paper’s motivation of analyzing classic prophet inequality’s robustness to strategic rewards, and could be suitable for a future work focusing on developing new prophet inequalities under strategically revealed reward signals.
>
> **1-1/e competitive ratio for iid case**
> The 0.745 competitive ratio for the iid case in the non-strategic setting is achieved by an **adaptive** threshold policy where the threshold for each player can depend on the previous reward realizations. This policy, however, is beyond the scope of this work as we focus on understanding how the classic static threshold policy performs (in both non-strategic and strategic settings). This is not to suggest that a dynamic threshold is uninteresting, but we aimed to maintain coherence in our results and avoid overwhelming readers with too many different settings. This is especially important as this study is the first of its kind, and the static threshold setting, while basic, is already quite intriguing.
> The 1-1/e competitive ratio, to our best knowledge, can be indeed achieved by a static threshold policy but requires some careful probabilistic tie-breaking rule when the searcher faces a tie between the realized reward and the stopping threshold (e.g., https://arxiv.org/pdf/2108.12893). While in our paper, we focus on a deterministic tie-breaking rule where the searcher always accepts the player if the expected realized reward is no smaller than the stopping threshold. The rationale behind this focus is that: (1) this class of static threshold policy with this simple tie-breaking rule are indeed effective policies for the classic prophet inequality, and we are studying their robustness in this paper; (2) they are more straightforward to implement in real-life cases.
> Lastly, we also kindly remark that in Corollary 5.3, we show there exists a static threshold policy that is 1-1/e approximation under the strategic setting (and this ratio is tight), however such policy has a bad performance in the non-strategic setting. Notice that our Proposition 5.2 rules out the possibility for all possible static threshold stopping policies (being within the spectrum in Definition 2.2 or beyond) to achieve $\alpha$-robustness with $\alpha>1/2$.

---

> ### Comment · Reviewer_vri4 · 2024-08-07
>
> I thank the authors for their response.
>
> * **Clarification of the searcher’s stopping policy.** I thank them for the clarification.
> * **Benchmarking within the threshold stopping policies** I agree that studying algorithms with adaptive thresholds is highly challenging. However, in many variants of prophet inequalities, when the observation order is fixed, as in the current paper, static threshold policies are enough to achieve an optimal competitive ratio. I am curious to know if this is the case also in the current problem or if adaptive thresholds can yield better competitive ratios. My question is not about studying general threshold policies, but instead on proving the optimality, or not, of the proposed static threshold policy among all algorithms, or at least among all algorithms with static thresholds (beyond the limited spectrum of Definition 2.2).
> * **1-1/e competitive ratio for iid case** The competitive ratio of $1-1/e$ in the iid case is achieved by a simple static threshold algorithm, which does not seem to require a random tie-breaking rule, see the proof of Theorem 3.1 in https://arxiv.org/pdf/2205.05519. I still believe that in the iid case, it makes much more sense to require a competitive ratio of 1-1/e in the non-strategic setting instead of 1/2.
>
> The authors' response addressed some of my concerns, though not all. However, in light of the other reviews and all the authors' responses, I have slightly raised my score to lean towards acceptance.

---

> > ### Author Response · Authors · 2024-08-11
> > **Response to Reviewer's Comments**
> >
> > We sincerely thank the reviewer for engaging in our discussions, and thanks for the insightful comments. We believe the following responses should help to address most of the reviewer’s concerns raised in Item 2 and 3 of the follow-up.
> >
> > **Benchmarking within the threshold stopping policies**:
> > We agree with the reviewer that understanding optimality of our threshold policy compared to best **dynamic policy** is a very interesting future direction. Our current paper did not study this question. However, when compared with optimal **static threshold** (not necessarily within the $1/2$-approximation threshold spectrum as in Definition 2.2), our $1/2$-robustness in Theorem 5.1 for IID case is indeed optimal. This is shown by our Proposition 5.2. (please also refer to its refined proof in our additional response to the Reviewer **mXL5**). It constructs examples to show that, even in IID case, **any** static threshold (not necessarily within Def 2.2’s spectrum) achieving at least $1/2$-competitive ratio for non-strategic setting cannot be $(1/2+\epsilon)$-robust for any $\epsilon > 0$.
> >
> >
> > **1-1/e competitive ratio for i.i.d. case**:
> >
> > 1. The referred paper requires distributions to have NO point mass (see page 5, the third line after the **Query** paragraph, for the assumption statement; essentially they need the threshold T such that $\text{CDF}(T) = 1-1/n$ to exist). In our setting, we strived to make least assumptions on the reward distributions to understand its robustness against the strategic reward signaling, thus we do not make any assumptions on the reward distributions and allow them to have point masses.
> > 2. Take a step back, even for the IID continuous distributions, it is not difficult to identify examples to show that the above threshold T with $\text{CDF}(T) = 1-1/n$ — which to the best of our knowledge is the only known threshold to achieve the $(1-1/e)$ competitive ratio for non-strategic setting  —  can become arbitrarily bad in strategic setting (i.e., cannot guarantee $\epsilon$-robustness for any $\epsilon > 0$). Reviewer **mXL5** happened to also ask this question, hence please refer to our last response to Reviewer **mXL5** for a construction of such an example.
> > 3. It is an interesting question to study whether there are other thresholds, other than the  $T = F^{-1}(1-1/N)$ (here F denotes the CDF) as in the referred paper, that guarantees $(1-1/e)$-CR in both non-strategic and strategic settings. However, this question is beyond the scope of this paper since it is not even known whether there is another threshold other than $T = F^{-1}(1-1/N)$ that can have  $(1-1/e)$-CR even just for the non-strategic setting.

---

> > > ### Author Response · Authors · 2024-08-12
> > >
> > > We sincerely thank the reviewer again for engaging in our discussions and hope that our response has effectively addressed your further questions. If there are any lingering questions, we would be more than happy to address them.

---

> ### Comment · Reviewer_vri4 · 2024-08-12
>
> I thank the authors for all the clarifications. I strongly encourage them to include the additional discussion of the IID case, and state more explicitly the limitations of their work indicated during the rebuttal in the revised version of the paper.
>
> The authors have addressed most of my concerns. I raised my score to reflect my satisfaction with their responses.

---

> > ### Author Response · Authors · 2024-08-12
> >
> > We greatly thank the reviewer for the feedback! We will make sure to make our results for the IID case more complete, and state the limitations of our work more explicitly in the revision.

---

### Official Review · Reviewer_mXL5 · 2024-07-15

**Soundness:** 3
**Presentation:** 3
**Contribution:** 3
**Rating:** 7
**Confidence:** 4

**Summary:**

This paper considers a bayesian persuasion variant of the classical prophet inequality problem: an online decision-maker will face a sequence of independent positive random variables $(X_1,\dots,X_n)$, $X_i \sim F_i$ known, and must decide when to stop in order to maximize the expectation of the selected item. Contrarily to the original problem,  each individual $i$ is strategic and chooses a signaling schemes aiming to maximize its chance of being selected (such as $\mathbb{1}_{X_i \geq t}$): the decision maker will only observe the signal from individual $i$, and not necessarily the $X_i$ directly. The process is in $3$ steps, the decision maker selects a stopping threshold $T$, the individuals select their signaling schemes, and finally the sequence of signals is observed. They first show that the usual rule of $\mathbb{E}[\max_i X_i]/2$ which without strategic agents achieves a $1/2$ competitive ratio, achieves a $(1-1/e)/2$ competitive ratio with strategic agents. In addition this is tight over a range of thresholds. Additional settings when the distributions are i.i.d. or are log-concave are considered.

**Strengths:**

1) This is a novel model which incorporates strategic behavior into prophet inequalities, adding to the growing literature of algorithms in strategic environments.
2) The paper is clear and well written.
3) The new results are significant, and non-trivial. Upper and lower bounds on the competitive ratio in the strategic setting are provided. This work also considers multiple sub-settings, giving more insight and leaving specific interesting open questions for this new problem.

**Weaknesses:**

1) It remains unknown whether $(1-1/e)/2$ is the best competitive ratio achievable in the strategic setting.
2) Some of the additional assumptions made when considering log-concave distributions are strong and too specific.

**Questions:**

Questions and remarks :
1) In the i.i.d. setting, the negative results do not seem to preclude the existence of a threshold with a $1-1/e$ CR in the non-strategic setting and a $1/2$ CR in the strategic setting. Is it possible to achieve? For instance, what is the performance of the quantile rule $T=F^{-1}(1-1/n)$ which achieves $1-1/e$ in the non-strategic iid setting?
2) Related to Corollary 5.3, do the authors know if better results are achievable in the non-iid strategic setting if we give up the performance in the non-strategic setting ?
3) Do the authors have any insights about the prophet secretary case?
4) I think it would be useful to give more details with respect to [29] as it seems to directly pertain to the topic at hand, and this could provide some insights in terms of what is gained in a centralized vs decentralized setting.
5) Regarding the payoffs obtained by the individuals when their item is selected, have other types of payoff structures been considered? For instance, player $i$ receives reward $X_i$ if item $i$ is selected, and $0$ otherwise. How does it affect the strategizing?
6) A brief comment on the existence of $T^{\dagger}$ and $T^*$ that satisfy their relevant equations should be added.
7) Are there examples of log-concave distributions which does not respect the additional slow decrease condition and which achieves a CR smaller than $1/2$ with the suggested range of thresholds? More broadly, are the additional assumptions (other than log-concave) truly necessary?

line 145: the \nicefrac{} command for $\mathbb{E}[\max_i X_i]/2$ makes it too small.

**Limitations:**

Some limitations regarding strong assumptions and tightness of results have been accordingly addressed in the paper.

---

> ### Author Rebuttal · Authors · 2024-08-06
>
> We sincerely thank the reviewer for the positive feedback and valuable comments/suggestions! Please find our response to each of your questions below.
>
> **Q1: Results for i.i.d case**
> We would like to first clarify that, to the best of our knowledge, the existence of a static threshold policy that achieves $1-1/e$ CR in non-strategic IID setting requires that the searcher carefully breaks the ties when she faces a tie between the realized reward and the stopping threshold (i.e. it involves some probabilistic tie-breaking, especially when facing distributions with masses). While in our setting, we consider a static threshold policy where the searcher always accepts the player when the expected realized reward is no smaller than the stopping threshold (see Definition 2.1). Thus, our negative result in Proposition 5.2 indeed excludes the case mentioned by the reviewer.
>
> However, we note that the reviewer might be asking that if there exists a static threshold that achieves an ½ CR in non-strategic setting and an 1-1/e CR in strategic setting. The answer to this question is also no: consider the example of $5$ iid distributions with binary support $\{4.2,s\}$ where $s$ is sufficiently large and the mean is $5$. One can compute in this case that $\lim_{s \to \infty} \text{OPT} = \lim_{s \to \infty} (\frac{s-5}{s-4.2})^5 \cdot 4.2 + (1 - (\frac{s-5}{s-4.2})^5) \cdot s = 8.2$. We first assume such a threshold $T$ exists. For the non-strategic setting, if $T > 4.2$, then the searcher only accepts realized reward $s$, so the searcher’s payoff is $\lim_{s \to \infty} (1 - (\frac{s-5}{s-4.2})^5) \cdot s = 4$. Clearly $4/8.2 < ½$ so we must have $T \le 4.2$. Then in the strategic setting, if $T \le 4.2$, then the searcher would always accept the first player, so her payoff is $5$ with clearly $\frac{5}{8.2} < 1-1/e$, showing that such a $T$ ceases to exist.
> This example also shows that $T=F^{-1}(1-1/n)$ does not achieve $½$ CR in a non-strategic setting. In fact, if we consider the same example of $5$ iid distributions on binary support $\{t,s\}$ but letting $t \to 5$ instead of $t = 4.2$, we can see that since $T > t$, the searcher only accepts reward $s$ in the non-strategic setting. If $t$ becomes close enough to $5$ and $s$ is large enough we see that the non-strategic CR for such $T$ can actually approach zero.
>
> However, we note that extending our results to more general stopping policies would be definitely an interesting future direction. We will add corresponding discussions in the revision.
>
> **Q2: Better results in strategic setting while ignoring performance in non-strategic setting**
> When ignoring the performance in non-strategic setting, in the paper we show that the best approximation ratio that a static threshold stopping policy can achieve is $(1-1/e)/2$. So far it is unclear to us if this ratio could be improved for the static threshold stopping policy.  Indeed, Reviewer 1242 pointed out a hard example provided in [19], under the same example we can conclude that the best approximation ratio of a static threshold policy one can achieve is no more than 0.4823 when players are strategic. However, whether $(1-1/e)/2$ is the optimal or if it can be improved for static threshold stopping policies still remains an open question, and we consider it as an intriguing future work.
>
> **Q3: Prophet secretary**
> Thanks for bringing out this interesting extension. We think the results we establish in Section 3 (i.e. characterizing the player’s optimal information revealing strategy when the searcher adopts a threshold stopping policy) still hold. Characterizing strategic-robust stopping policy may require significant effort for this problem and we leave it as another interesting future direction.
>
> **Q4: Comparisons to [29]**
> We thank the reviewer for this suggestion. We agree with the reviewer and will add more discussions about the comparison to the work [29].
>
> **Q5: Different payoff structure**
> Our results can be easily extended to the setting where each player $i$ has a payoff $v_i \ge 0$ if his reward is accepted and 0 otherwise (please see our footnote 6 for this discussion). Using the terminology in information design, here the player’s payoff is essentially state-independent (i.e., it only depends on the searcher’s action and not on the realized reward $X_i$). It would be interesting to explore how the results would change if the player’s payoff is state-dependent.
>
> **Q6: Existence of** $T^\dagger$ **and** $T^*$
> We thank the reviewer for this suggestion. We will add a comment about the existence of $T^\dagger$ and $T^*$. The proofs in the main text are for the continuous distributions, thus $T^\dagger$ and $T^*$ always exists. The definitions of $T^\dagger$ and $T^*$ are slightly different for the distributions with atoms, and they also always exist.
>
> **Q7: Additional assumptions for log-concave distributions**
> We acknowledge that, after extensive simulations, we did not find an instance with log-concave distributions that has strictly smaller than ½-robustness when this instance does not satisfy the given conditions. These assumptions are meant to serve as smoothing parameters to imitate nice behavior of endpoints on [0,1]. We believe relaxing these assumptions may require significant effort and we leave it as an intriguing future work.

---

> > ### Comment · Reviewer_mXL5 · 2024-08-09
> >
> > We thank the authors for their responses and clarifications. I still have a few questions left,
> >
> > Q1. I agree that when the distributions are discrete a tie-break is required to achieve $1-1/e$. However much of the paper’s main section presentation assumes continuous distributions, in which case the $1-1/e$ competitive ratio is achieved. I would expect that under continuous distributions the natural definition for i.i.d. $\alpha$-robustness would be to guarantee the non-strategic competitive ratio of $1-1/e$. Even without the continuous assumption, does allowing for random tie-breaks make any of the proofs catastrophically fail? If this is the case this should be pointed out, as allowing for tie-break is quite standard in the literature, in which case looking at thresholds without tie breaks might not be the right policy set to consider.
> >
> > I am also now confused about the current proof of Proposition 5.2 in the paper. Is it correct that what the counter example proves, is that any fixed threshold (without tie breaking) in the non strategic setting will bet at most $½$? Maybe I misunderstood something, but this does not prove that for any $\alpha$-robust policy, $\alpha \leq ½ $, as we could still have $½$ in the non strategic setting and more than $½$ in the strategic one.
> > However your new counter example does show that if $½$ is achieved in the non strategic setting, then the competitive ratio is at most $5 / 8.2$ in the strategic setting, but this is still not $½$. Please feel free to correct me.
> >
> > Regarding $Q7$, I think this limitation should therefore be included next to the result for transparency.

---

> > > ### Author Response · Authors · 2024-08-11
> > > **Response to Reviewer's Comments**
> > >
> > > We appreciate the reviewer’s engagement and very interesting questions, which also help us to further deepen our results. Our response below is a bit long, but it should resolve the reviewer’s three major questions we believe. In a nutshell,
> > > 1. We identify an example to show that, even for continuous distributions, the known threshold to achieve the (1-1/e) competitive ratio (CR) in IID case for non-strategic setting can become arbitrarily bad in strategic setting (i.e., cannot guarantee $\epsilon$-robustness for any $\epsilon > 0$). While this is a very interesting question, it is unclear whether one could obtain interesting $\alpha$-robustness results for IID setting, subject to $(1-1/e)$-CR in non-strategic situations. Answering this question will also require significant advances over the state-of-the-art results for standard (non-strategic)  prophet inequality for IID cases, as the currently known threshold would not work.
> > > 2. We thank the reviewer for pointing out the incompleteness of current proof of Proposition 5.2 to our attention. We acknowledge that the current proof of Proposition 5.2 indeed does not fully prove the statement, though it has the right intuition and ideas (and the statement of Prop 5.2 is also correct). A slight modification of the constructed example by changing $N-1$ in the construction to $N-\alpha_1$ and then letting $\alpha_1 \to 1$ suffices to prove the proposition. We also include a complete argument below in case the reviewer would like to take a look.
> > > 3. Lastly, we'd also like to note that tiebreaks do not affect our proofs. In particular, allowing tiebreaks does not help to get a better CR in a strategic setting. This is followed by a simple observation of the structure of the players’ optimal information strategies (see Proposition 3.1), where each player’s optimal information strategy is a two-point-mass distribution with larger mass being the same $T$. Thus, the searcher would never be better off by randomly rejecting any player if he is realizing a mass $T$ since that is the maximum possible value sent by all the players.
> > >
> > > Needless to say, we will incorporate all the above discussions in the next draft and also follow the reviewer’s suggestion and add our discussions about $Q_7$.
> > >
> > >
> > > **The threshold with $1-1/e$ competitive ratio in IID case is NOT $\epsilon$-robust for any constant $\epsilon>0$**:
> > > Under **continuous** iid distributions, one can always achieve $1-1/e$ competitive ratio (CR) in the non-strategic setting using the threshold $T$ such that $T = F^{-1}(1-1/N)$, where $F$ is the CDF. An interesting question is whether this $T$ is $(1-1/e)$-robust in the sense that it also achieves $1-1/e$ CR in the strategic setting. Unfortunately, the answer to this question is no --- in fact, this threshold $T$ cannot guarantee $\alpha$-robustness for any positive $\alpha$.
> > >
> > > We provide an intuitive counterexample below, but happy to convert it to a rigorous construction (which should require no new ideas but just tedious math constructions) if the reviewer is interested to see. Notably, it is an interesting question to study whether there are other thresholds, other than the threshold $T = F^{-1}(1-1/N)$, that guarantees $(1-1/e)$-CR in both non-strategic and strategic settings. However, this question is far beyond the scope of this paper  since, to our knowledge, it is even not known whether there is another threshold other than $T = F^{-1}(1-1/N)$ that can have  $(1-1/e)$-CR even just for the non-strategic setting.
> > >
> > > The counterexample considers $N$ players with continuous iid distributions that almost have a binary support in $\{0,1\}$, with probability $ \frac{1}{2N}$ to be $1$ hence mean $1/(2N)$ (rigorous construction only needs to ``smooth'' this discrete distribution to be a continuous one that mostly concentrates its mass on $0, 1$).  These continuous distributions can be made to satisfy $F^{-1}(1-1/N) < \frac{1}{2N}$ (i.e., the threshold is very close to $0$ but just a little bit larger).
> > >
> > > Note under these distributions, the first player can use the no-information strategy and reveal $\delta(\frac{1}{2N})$ and will always be picked by the searcher. Hence the searcher’s payoff is $u^s(T) = \frac{1}{2N}$. Since we can approximate continuous distributions close to the binary support one, for any $\epsilon > 0$ we can always find a continuous distribution such that its $\text{OPT} = (1 - (1-\frac{1}{2N})^N) - \epsilon$, as $(1 - (1-\frac{1}{2N})^N)$ is the $\text{OPT}$ of the binary support one. Then the searcher’s CR is $\frac{u^S(T)}{\text{OPT}} = \frac{1}{N \cdot (1 - (1-\frac{1}{2N})^N - \epsilon)}$, which goes to $0$ as $N \to \infty$. Hence the searcher’s CR in the strategic setting can get very close to $0$, showing that $T = F^{-1}(1-1/N)$ is not robust to strategic reward signaling.

---

> > > > ### Author Response · Authors · 2024-08-11
> > > > **Response to Reviewer's Comments (Continued)**
> > > >
> > > > **(Refined) Complete Proof of Proposition 5.2**:
> > > >
> > > > Consider an instance with $N$ iid distributions on binary support $\{N-\alpha_1, N + \alpha_2\}$ for $ \alpha_1 \in (0, 1), \alpha_2 > 0$ such that the probabilities are chosen to make the mean of each distribution precisely $N$ -- i.e., the probability of taking the smaller value is $\frac{\alpha_2}{\alpha_1 + \alpha_2}$. For the purpose of this proof, one should think of $N, \alpha_2$ as being very large. The prophet value $\text{OPT}$ can be calculated as following:
> > > > $$\text{OPT} = (N +  \alpha_2) \cdot  [1 - (\frac{\alpha_2}{\alpha_2+\alpha_1})^N]  + (N - \alpha_1) \cdot (\frac{ \alpha_2}{\alpha_2 + \alpha_1})^N.$$
> > > >
> > > > We first claim that, for *any* $\alpha_1 \in(0, 1)$, no threshold $T \in (N - \alpha_1,  N + \alpha_2]$ can guarantee at least $1/2$ competitive ratio for non-strategic settings. Intuitively, this is because any such threshold will give up any realization of $N- \alpha_1$ value, which leads to bad performance when the probability of having at least one realization of $ N + \alpha_2$ is small (i.e., when $\alpha_2$ is very large). Detailed calculation is as below: any threshold $T \in (N - \alpha_1 , N + \alpha_2] $ only accepts the realized $N + \alpha_2$ value, hence the expected payoff for non-strategic (ns) setting is (i.e., the first term of $\text{OPT}$) $$u^{ns}(T) = (N + \alpha_2)  \cdot [ 1 - (\frac{\alpha_2 }{\alpha_2 + \alpha_1 })^N].$$
> > > >
> > > > Therefore,
> > > > \begin{eqnarray*}
> > > >     \frac{u^{ns}(T)}{\text{OPT}}  &=&  \frac{ (N + \alpha_2)  \cdot [ 1 - (\frac{\alpha_2 }{\alpha_2 + \alpha_1 })^N]  }{ (N + \alpha_2)  \cdot [ 1 - (\frac{\alpha_2 }{\alpha_2 + \alpha_1 })^N]  + (N - \alpha_1) \cdot (\frac{ \alpha_2}{\alpha_2 + \alpha_1})^N  }  \\
> > > >     &=&  \frac{  1  }{ 1   + \frac{N - \alpha_1}{N + \alpha_2} \cdot \frac{(\frac{ \alpha_2}{\alpha_2 + \alpha_1})^N}{   1 - (\frac{\alpha_2 }{\alpha_2 + \alpha_1 })^N   }  }
> > > >     \end{eqnarray*}
> > > > We now analyze the limit of the key term $    \frac{N - \alpha_1}{N + \alpha_2} \cdot \frac{(  \frac{ \alpha_2}{\alpha_2 + \alpha_1})^N}{   1 - (\frac{\alpha_2 }{\alpha_2 + \alpha_1 })^N   }   $. To ease the analysis, we re-parameterize the above term with $\alpha_2 = \frac{\alpha_1 N }{ \alpha}$ with free parameter $\alpha$ to obtain
> > > > \begin{eqnarray*}
> > > >   \frac{N - \alpha_1}{N + \alpha_2} \cdot \frac{( \frac{ \alpha_2}{\alpha_2 + \alpha_1})^N}{   1 - (\frac{\alpha_2 }{\alpha_2 + \alpha_1 })^N   }     =     \frac{N - \alpha_1}{N} \frac{1}{1 + \alpha_1/\alpha}  \cdot \frac{(  \frac{ N }{N + \alpha})^N}{   1 - (\frac{ N }{N + \alpha })^N   }  = A(N, \alpha; \alpha_1)
> > > > \end{eqnarray*}
> > > > Since $\alpha_1 \in (0, 1)$ is fixed. Now fix $\alpha$ and let $N \to \infty$, we have $\lim_{N \to \infty} A(N, \alpha; \alpha_1) =    \frac{1}{1 + \alpha_1/\alpha} \cdot \frac{e^{-\alpha} }{   1 - e^{-\alpha} }$. Now, letting $\alpha \to 0$ and applying standard l'hopital's rule, we have  $\lim_{ \alpha \to 0 }   \frac{1}{1 + \alpha_1/\alpha} \cdot \frac{e^{-\alpha} }{   1 - e^{-\alpha}  }  =  \lim_{ \alpha \to 0 }   \frac{\alpha e^{-\alpha} }{(\alpha + \alpha_1)(1 - e^{-\alpha} ) }  = 1/\alpha_1$. As a consequence, we have $\lim_{N \to \infty, \alpha_2 \to \infty}   \frac{u^{ns}(T)}{\text{OPT}} = \frac{\alpha_1}{\alpha_1  + 1}$. Thus, for any $\alpha_1 \in (0, 1)$, there exists an IID instance with large enough $N$ and $ \alpha_2 $ such that no threshold $T \in (N-\alpha_1, N +  \alpha_2]$ can achieve CR at least $1/2$.
> > > >
> > > > The above analysis shows that the only threshold $T$ that can guarantee a CR at least $1/2$ in the above class of instances must satisfy $T \le N-\alpha_1$.
> > > > Any such threshold will achieve searcher's utility $u^s(T) = N$ in the strategic setting since all players will reveal no reward information, leading to expected reward value $N$ for each player.
> > > > A similar limit analysis using the same reparameterization $\alpha_2 = \frac{\alpha_1 N }{ \alpha}$ shows that  $ \lim_{\alpha \to 0 }\lim_{N  \to \infty}  \frac{u^{s}(T)}{\text{OPT}} = \frac{1}{1+\alpha_1}$.
> > > >
> > > > The arguments above shows that for any $\alpha_1 < 1$, as $N, \alpha_2 \to \infty$ in the above instances,  any threshold that has CR at least $1/2$ in non-strategic setting will have CR tending to $ \frac{1}{1+\alpha_1}$ in the strategic setting. Since this conclusion holds for any $\alpha_1 \in (0, 1)$, it rules out the possibility of  having an $(1/2+\alpha)$-robust threshold (even outside the $1/2$-approximation threshold spectrum as in Definition 2.2) for any  $\alpha > 0$.

---

> > > > > ### Author Response · Authors · 2024-08-12
> > > > >
> > > > > We sincerely thank the reviewer again for engaging in our discussions and hope that our response has effectively addressed your further questions. If there are any lingering questions, we would be more than happy to address them.

---

> > > > > > ### Comment · Reviewer_mXL5 · 2024-08-13
> > > > > >
> > > > > > I thank the authors for replying to my comments and have no further questions. I think after taking into account the suggestions/comments made by the reviewers and authors this will be a nice paper.
> > > > > >
> > > > > > Best,

---

> > > > > > > ### Author Response · Authors · 2024-08-13
> > > > > > >
> > > > > > > We greatly thank the reviewer for the feedback and encouraging words. Indeed, the comments/suggestions made by the reviewer (and also other reviewers) would be helpful to improve our paper and deepen our results. We will make sure to include these suggestions/comments in our revision.

---

### Decision · Program_Chairs · 2024-09-25

**Decision:**

Accept (poster)

**Comment:**

The paper investigates the robustness of the so-called Prophet inequality to strategic reward signaling. This inequality concerns simple threshold policies (the decision maker accepts the reward if it exceeds a threshold). Now the authors consider a scenario where the successive rewards are revealed by selfish agents willing to maximize the probability that their reward is selected. They show that if the decision maker applies a threshold-based policy applied to the posterior mean rewards given the signals sent by the players, then the optimal strategy of the players are simple binary policies.
Based on this result, the authors first establish that the KW threshold achieves a $(1-1/e)/2$ approximation of the OPT under strategic players (Theorem 4.1). The result is extended to the particular case of IID rewards and log-concave distributions (with a ½ approximation ratio).

Based on the reviews, the discussions during the rebuttal phase, and my own evaluation, we encourage the authors to modify their manuscript to account for the following points.

1. Clarity of the model. The game-theoretical model is described at the end of Section 2, and as pointed out by two reviewers, it needs to be really clarified.
(i) The decision maker. She knows the distributions $H_i$ and selects a threshold accordingly. The threshold will be comapred against the posterior means of the rewards given the signals sent by the players.
(ii) The players. They know the threshold and know how it will be used by the decision maker.
The objective is for players to design optimal signaling strategies, and to characterize the competitive ratio of the corresponding Nash Equilibrium (in terms of the selected reward). Currently the way the problem is presented is confusing.

2. Technical precision and proofs. Some of the reviewers (e.g. mXL5) asks for more clarity in the proofs. Also please state results rigorously in the main text (Prop. 3.1 is confusing because you do not specify the optimal strategy in the case where the threshold is greater than the upper bound of the reward distribution support – this is done properly in Appendix).

3. Adding discussions. Please include the discussions that you had with the reviewers. I would specifically recommend discussing the limitations of the model. One limitation pertains to the fact that the searcher is assumed to apply a threshold policy based on the posterior means of the rewards given the signals sent by the players. This implies that the players send both the signal $\sigma_i$ sampled from $\phi_i(.|X_i)$ ‘and’ $\phi_i$. The theory works because we assume that the players are truthful in the sense that they report the sample $\sigma_i$ actually sampled from $\phi_i(.|X_i)$. Could the authors discuss what societal mechanisms would encourage or enforce truthfulness, so that this is a reasonable, albeit idealized, assumption?